# Living Images and Marian Devotion: Words, Gestures, and Gazes

**Fuensanta Murcia Nicolás** 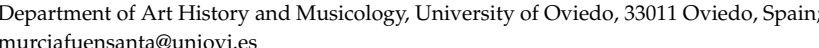

Department of Art History and Musicology, University of Oviedo, 33011 Oviedo, Spain;
murciafuensanta@uniovi.es

**Abstract:** This article examines the living images of the Virgin through the illustration of one of the most important collections of miracles of the 13th century, *Les Miracles de Nostre Dame* by Gautier de Coinci. In this case, I will focus my attention on manuscript 551 of Besançon (Besançon, BM, MS 551), which, although it has many flaws in its manufacture, offers an interesting presentation of living images. The study of these miniatures reflects the importance of devotion, the set of gestures, words, and gazes, in the medieval spectator's experience of Marian images. At a time when these images' legitimacy as sacred objects was still being debated, the artists in this manuscript show their power without censorship, presenting them as if they were the Virgin herself.

**Keywords:** Marian miracles; Gautier de Coinci; Illuminated Manuscripts; miraculous images; medieval visual culture

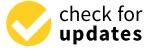

## 1. Introduction

> In so many places the Mother of God works so many miracles and wonders that the whole world marvels at them[1].

Cult images in the Middle Ages, especially those of a miraculous nature, have occupied an important part of Visual Culture Studies over the last two decades. In the first centuries of the Middle Ages, Christian intellectuals rejected their sacred character, giving them a didactic function (Chazelle 1990, 1995; Schmitt 2002, pp. 63–95; Wirth 2001). However, the very evolution of liturgy and devotion meant that this role acquired other nuances, such as stimulating spiritual contemplation and recalling the most important events in Sacred History (Jung 2010; Kessler 2006b; Palazzo 2010). This circumstance, which did not limit individual experience, meant that the boundaries with veneration became increasingly blurred, and, with the recovery of Byzantine postulates from the 12th century onwards, Western authors gradually constructed a discourse that legitimized their position as sacred objects (Boulnois 2008, pp. 237–42; Kessler 2006a). It would be in the following century when the semantics and ontology of the images were equated, and, under the protection of the new frameworks of thought, it was recognized that the *virtus sancta*, the divine power, could work miracles through them (García Avilés 2010; Sansterre 2009b).

Apart from theoretical debates, attention has also been paid to other types of sources, further away from intellectual debates, which focus on the devotion they aroused in the viewer. In this field, the most illuminating cases are the stories about images that prove to have miraculous qualities (Belting 2009, pp. 407–9; Freedberg 2009, pp. 123–25; Sansterre 2013). In contrast to the previous debates, here we do not find so many discrepancies; quite the contrary, that power makes them objects of worship from an early date (Chazelle 2005; Sansterre 1998, 2020, pp. 21–35). Comparing the development of both discourses, theological and devotional, we see that they run parallel to each other until the 13th century, when they finally converge. This is largely due to the fact that the two positions have a point in common, the miracle (Brown 1989, p. 329; Ward 1982).

Within this large group we also find differences. The images of the Crucified are the first to appear as miraculous (Horn Fuglesang 2004; Palazzo 1992; Sansterre 2005, 2009a), even before the year 1000 and the resurgence of monumental sculpture, while the Marian images appear mainly during the 12th century (Gold 1985; Marks 2004; Russo 1996; Sansterre 2006). This discrepancy is due to the construction of the character of the Virgin and the development of her cult. Although her image was already present in the liturgical context, given that she symbolized the Incarnation, the increase in the number of accounts is due to the consolidation of her role as mediator, which had been crystallizing since the 11th century (Clayton 1990, p. 77; Fulton 2002, pp. 218–21; Oakes 2008; Rubin 2009a, p. 27). This role made her more independent from her strictly maternal condition, which was reflected in the rise of Marian pilgrimage centers and the writing of the first collections of miracles (Albert-Llorca 2002; Bayo 2004; Fuchs 2006, p. 69; Rubin 2009a, pp. 185–87; Signori 1996). Although the latter were intended as a compilation, they were the seed of the works that would first appear in the 13th century and which would be considered a literary genre in their own right (Montoya Martínez 1981). These new compilations would no longer be written in Latin but in vernacular, would have a defined internal structure, and would be intended to honor the Virgin and to teach readers about her cult and that of her images (Sansterre 2010; Murcia Nicolás 2016, pp. 22–27).

Among these works, the Benedictine Gautier de Coinci's *Les Miracles de Nostre Dame*, written before 1236, stands out. The final version, which went through several stages of redaction (Okubo 2005), consists of two volumes with an identical table of contents: it begins with a prologue, where the author makes his motivation clear, followed by a group of songs, and continues with the narration of the miracles in verse and several lyrical compositions (Ducrot-Granderye 1932, p. 169; Grossel 2001). In addition, and to further highlight their sobering character, he ends each story with a reflection for the reader which, although it is a moral drawn from the events narrated, ultimately emphasizes the role of the Virgin as an all-powerful mediator (Benoit 2007; Grossel 2005; Montoya Martínez 1979). *Les Miracles de Nostre Dame* was widely distributed if we consider the number of manuscripts preserved, more than fifteen with the complete version of the text, many of which are also illustrated (Duys et al. 2006). This particularity makes this corpus ideal for the study of miraculous images, since both the textual and the visual aspects come together.

In the second half of the 13th century, six examples are illustrated, three of which are linked to the same workshop located in the Soissons area, two two the Paris region, and a sixth possibly made in the southern half of France (Stones 2006a). The latter, preserved in the Besançon Library, is the most controversial. While the rest are finished, which has allowed for a more extensive study of their codicological and iconographic features and even the establishment of relationships with other later copies, the Besançon manuscript 551 is incomplete, only has the miniatures of the first part, and contains many errors and lacunae in its text. When we analyze its possible production process, we find two teams with different qualities: the first, present in the first notebooks, is far superior to the second, as its miniatures are much more complex and elaborate (Stones 2006b, pp. 93–95; Murcia Nicolás 2013, 2014). It is plausible to think that it was commissioned as a luxurious manuscript, but, for reasons unknown to us, its quality declined and it was never finished. The lack of documentation, coupled with its uneven manufacturing process, has led to its dating being disputed and revised. At first it was catalogued as an example made in the first decades of the 14th century, but later, according to codicological criteria, it was placed, with reservations, between the 1260s and the 1270s (Stones 2015, p. 163; Russakoff 2019, p. 124). Despite its dubious chronology and its heterogeneous artistic quality, it is the manuscript with the most illustrations in the whole corpus. The other copies contain around eighty, whereas this has 136 finished illustrations, a number that more than doubles (over 300) if we take into account those that were projected.

In addition to this important number, which gives it a much superior visual narrative, there is the particular representation of the miraculous images (Murcia Nicolás 2012; Russakoff 2004). In the rest of the corpus, from the second half of the thirteenth century, we

find standards of representation that show iconographically the importance of the image in Marian miracles, either as mediators of the event or as sacred objects. However, in the case of those that come to life, the presentations are somewhat disparate (Murcia Nicolás 2016, p. 60). In some cases, there is an omission of the living character, possibly because of the misgivings that this attribution could still arouse (Camille 2000, p. 252). In contrast, in the Besançon manuscript 551, these images are shown uncensored and are even included without any textual reference. Until now, this subject has always been approached as a whole, comparing the different manuscripts and establishing the relationship between written and visual culture (Murcia Nicolás 2016, 2017, 2020; Russakoff 2019).

This manuscript is of particular interest because it allows us to study the new visual culture of the 13th century regarding these Marian images. Its miniatures are a reflection not only of Gautier de Coinci's text but also of the importance that devotional practices had acquired. The examples analysed demonstrate the sacred character of the image, which works miracles through the words, gestures, or looks addressed to it. Although the two illustrators of manuscript 551 have different styles, the idea they capture in their miniatures is the same: the living character is the maximum expression of the power of the image, which is amplified by the mediating and close character of the Virgin.

## 2. Love and Hate: The Power of Words

In the prologue to the second volume, Gautier de Coinci encourages his fellow ecclesiastics to devote all their love to the Virgin, since she is superior to any lady in the pastoral tales. This declaration, typical of courtly love, is present in a group of stories where the protagonist is admonished for not keeping a promise of a loving nature (Baum 1919). One of the best known, the so-called "miracle of the bridegroom", is about a young man who is betrothed to a statue of the Virgin, to whom he gives a ring as proof of his love. What is relevant in the text is his declaration, marvelling at the beauty of the image:

> One day they were playing ball in front of the doorway of a church, where there was a seated image ( . . . ) He looked at the image, which was new and fresh. When he saw its beauty, he knelt down in front of it, and devoutly bowed down and greeted it. In a short time, his will is transformed. "Madam," he says, "I will serve you from now on all my life, for I have never seen a young woman so beautiful and seductive in my eyes. You are a hundred thousand times more beautiful and lovely than the one who gave me this ring. I had given her my heart, but because of my love for you, I want to leave her with her affection and her jewels. I want to give you this beautiful ring for my sincere love, and I promise never to have any other friend or woman than you, sweet and beautiful lady". The ring he wears, he puts it on the finger of the image and, quickly, the statue closes it so tightly that no one could remove it without cutting it off[2].

As if it were a wedding ritual, the young man puts on the ring to sign his promise, which he later breaks, provoking the Virgin's anger at his infidelity:

> "You have acted neither correctly nor legally towards me," she says, "You have discredited yourself in front of me. Here is your friend's ring, which you gave me for your sincere love. You said I was a hundred thousand times more beautiful and seductive than any other young woman. You would have had in me a faithful friend if you had not forsaken me. You have left the rose for the nettle, the rose hip for the elder. You wretch! You are so credulous that you leave the fruit for the ear, the lamprey for the seven-eyed, for poison and gall you leave the sweet honeycomb and honey". The clergyman, stupefied by the vision, awoke with a heavy heart. He thought he would find the statue beside him[3].

Apart from the fact that we find a story from the classical world Christianized to become an example of renunciation and abstinence, the importance of the protagonist's declaration (Smith 2006, p. 168) is noteworthy. It is his words that provoke the response of the image, which closes its finger to preserve the proof of the love professed. In the

miniature in manuscript 551 we see the young man kneeling while the image extends his hand for him to place the ring on it (Figure 1). The illustrator has removed the references to the architectural context, since it is a statue placed in the doorway of a church, to make it an intimate and private scene. He has also stripped her of her status as an object, presenting her on the same level, face to face, with her admirer. Gautier de Coinci makes her share in the beauty of her model, the Virgin, and makes it clear that the words addressed to the former are as valid as if they were addressed to the latter in person (Rubin 2009b, p. 231; Sansterre 2010, p. 155). The same message is conveyed by the illustration, which, beyond showing the living character, humanizes the statue.

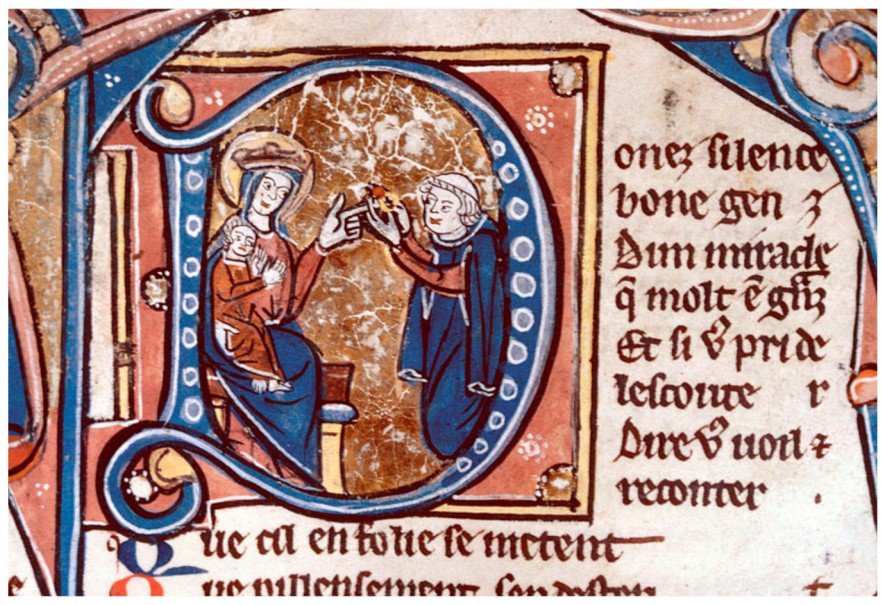

**Figure 1.** The young man betrothed to a statue. Gautier de Coinci, *Miracles de Nostre Dame*. Besançon, BM, MS 551, fols. 43r. Photo: Bibliothèque Municipale de Besançon.

Another miracle that demonstrates the power of the word as a trigger is that of the desecrated icon. A Jew visits the house of a Christian who has an image of the Virgin. The antagonist, who later ends up throwing it into a latrine, questions the veracity of its sacredness, clearly expressing his repulsion:

> Near him there was an alcove, and he looked and noticed an icon that displayed a small image that looked like Our Lady. "Tell me", he said, "by your soul, who is this image of?" "She is" the Christian replied, "of the Virgin, who is so pure, clean, and innocent, that the Lord of all people took on human form in her loins". The blood of the Jew boiled while speaking to him. "Do you venerate" he said, "Him whom we choose not to name? One should indeed beat you or tear your heart apart like a cow! You could just as well venerate an old pillar or a beam and bow before it and adore it as you do him of whom you speak to me. Fi!" said the dog [i.e., the Jew] "Too great is the shame, too great the outrage, too great is the affliction, that any man believes that the great God was born of this image of the Virgin Mary. There are no churches or even chapels, where there are not six or seven of these images. Such great shame should never happen!"[4].

The Jew profanes the image twice. The first occurs when he refutes the legitimacy of the image and the second when he rejects Mary's motherhood. His words will be punished accordingly, his tongue torn out and he himself led away by a group of demons:

> The Mother of God, who was in this image, did not want to suffer from this great outrage. Cruelly and quickly, she paid him back, because she struck him with a

piece of wood covered in mud. The Jew's tongue flew out of him. The Jew's soul and body taken all at once by evil spirits[5].

Gautier de Coinci explicitly cites the presence of the Virgin in his image, although the interpretation of who executes the punishment is doubtful. The illustration in manuscript 551 deviates from the usual depiction, focusing on profanation, to show retaliation (Murcia Nicolás 2016, pp. 59–60; Russakoff 2019, p. 37). However, the choice of illustrator is surprising (Figure 2). First, it is the image that pushes the Jew, but, more relevantly, it is how he represents it. Even though the title alludes to the term *yconia*, he opts for a sculptural representation. It seems clear that he wants to show the living character of the image, despite the dubious textual reference, and for this purpose the three-dimensional format is more suitable. Although in the Western tradition such images are mainly sculptural (Palazzo 2020, p. 114; Sansterre 2020, pp. 270–71), the illustrator has modified the interpretation to include an animated effigy.

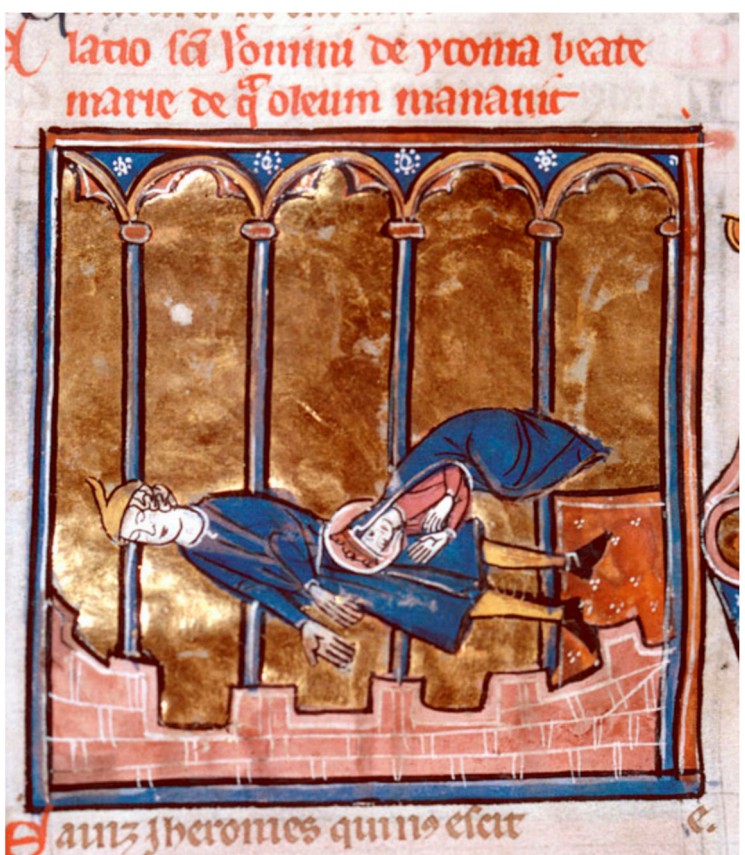

**Figure 2.** The Marian image insulted. Gautier de Coinci, *Miracles de Nostre Dame*. Besançon, BM, MS 551, fols. 32r. Photo: Bibliothèque Municipale de Besançon.

The two stories are based on different assumptions. The first responds to an idealization of the Virgin as a lady of courtly love, while the second demonstrates the transfer of an act associated especially with the Crucified, profanation, to other Christian representations (Bacci 2005, pp. 32–34; Sansterre 2013, p. 81; Schmitt 1998). However, they share a common element. The declaration made before the image has the same value as if it were made before the Virgin herself. The popularization of these stories of living images is instigated by the belief that the divine, the *virtus sancta*, can activate them and make them adopt human attitudes (Freedberg 2009, p. 339; García Avilés 2007; Vauchez 1999). From the end of the twelfth century, but especially in the thirteenth century, miracles gave greater independence to the Virgin, with her image, and not that of her son, coming to life (Barnay 1999, pp. 39–41; Fulton 2002, p. 218; Sansterre 2020, p. 274). This shift is mainly due to

two factors, the humanization of the sacred, which transforms Mary into a character close to human experience, and the development of private devotion towards closer spheres (Camille 2000, p. 243; Murcia Nicolás 2020). In this context, prayer becomes more emotional, to provoke greater proximity between the faithful and the sacred personage, and the image becomes the main addressee, as it is able to show the invisible in the visible world (García Avilés 2010; Kessler 2007; Palazzo 2010; Sansterre 2013). The final message is that words spoken in front of an image are heard and considered. That real presence is what we see in these two miniatures from manuscript 551: the Virgin responds through her animated effigy, either to claim a promise of love or to punish a declaration of hatred.

## 3. Honoring and Greeting: The Value of the Gesture

Gautier de Coinci makes numerous references to the importance of honoring images throughout the text (Murcia Nicolás 2016, p. 27; Sansterre 2010). This act consists of kneeling and saluting, after which, in gratitude, the Virgin acts in favor of the protagonist, using her image if necessary. In this sense, it tells the story of a nun who tries to flee from the convent to escape with her lover, an attempt that is thwarted on two occasions by the statue that presides over the chapel:

> As night falls on the community, the damsel discreetly leaves the dormitory. On the right she found the chapel dedicated to Our Lady, which she quickly entered. Her heart pounding, and as she used to do, she knelt in front of her image, which she humbly greeted. She got up quickly and went to the door, but the image that was inert, without delay, stood in the doorway with its arms crossed. It was so large that the young woman could not pass ( . . . ) The nun could not sleep, she got up from her bed and went to the chapel, for there was no other way out. Very humbly she knelt as she passed the altar and did with the image what she had done the first time. She is shocked when She sees her again at the door, blocking her way. She stretches out her arms in front of her face, as if to say, "Dear friend, you shall not pass this way"[6].

As in the miracle of the bridegroom, here the Virgin acts as the guarantor of abstinence and guardian of feminine virtues (Drzewicka 1985; Garnier 1985), preventing the nun from fleeing just after kneeling in front of her image. Genuflection is a gesture associated with the worship of images from before the year 1000 (Sansterre 2020, pp. 60–61), but those representing the Virgin would become more prominent from the 12th century onwards, thanks to the Marian sanctuaries (Sansterre 2010, pp. 56–58; Sansterre and Henriet 2009, pp. 64–65; Turner and Turner 1978, p. 171). In these centers, the image, seen as Mary's intermediary and capable of working her wonders, became the focus of prayers and greetings (Freedberg 2009, p. 119; Henriet 2006, p. 243). This type of devotional practice was legitimized by the revaluation of Byzantine ideas, in particular the concept of *transitus*. The fundamental basis of the cult of icons, it dictated that "the honor given to the image is received by its sacred prototype", so that images were not worshipped, but venerated to honor the personage they represented (Kessler 2006a, p. 155; Schmitt 2002, p. 90).

Throughout the 13th century there were misgivings about accepting that images could be venerated as sacred objects, but part of the theoretical discourse assumed the assumptions inherited from the work of John Damascene. One of the best-known examples is Thomas Aquinas' statement concerning the image of Christ:

> Therefore, we have to say that no veneration is to be paid to the image of Christ as an object, whether in carved or painted wood, because only the rational creature can be venerated. We show veneration only for what it represents. The same honor is to be paid to the image of Christ as to Christ Himself. Thus, since the cult of *latreia* is due to Christ, it is logical to do the same to his image[7].

The scenario set out by the Dominican saint also takes us back to the consequences of the Fourth Lateran Council. The Dogma of Transubstantiation, which recognized the presence of Christ in the Eucharistic substances, appears here applied to images, establishing

for them the same honor that is paid to their model (García Avilés 2010, p. 35). Gautier de Coinci does not reflect on these questions, but he does include the same conclusions in a devotional context (Murcia Nicolás 2016, p. 25). The inclusion of genuflection appears as a trigger for the miraculous event in the story of the nun. It is this gesture that makes it possible for the image to come to life to prevent her from fleeing, thus creating a cause-effect relationship, which we see in the miniature of this miracle (Figure 3). The illustrator chooses to embody the model in his own representation, like that seen in the example of the young man with the ring. The juxtaposition of the two scenes not only indicates that the Virgin was present, animating her statue, but also that the gestures of veneration are not in vain, as she receives them and acts accordingly.

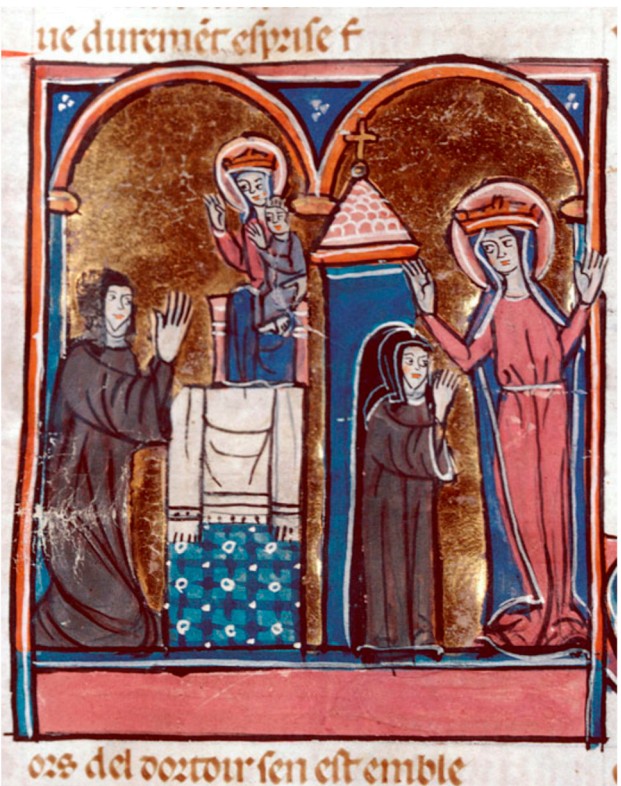

**Figure 3.** The nun who escapes from the convent. Gautier de Coinci, *Miracles de Nostre Dame*. Besançon, BM, MS 551, fol. 80r. Photo: Bibliothèque Municipale de Besançon.

Another aspect that helps us to understand these signs of devotion is the evolution that the Virgin undergoes. The Lateran synod also promoted her human motherhood, even above the spiritual (Bynum 2011, pp. 135–49; Gauthier 1993). Although her figure had always been linked to the incarnation of Christ, her images would become more realistic to show the bond with her son in a more empathetic and close way (Trotzig 2004). In manuscript 551 we find several miracles, the first miniature of which depicts the protagonist in front of a statue of the Virgin: the story of the noble woman of Rome (Figure 4), the pregnant abbess (Figure 5), and that of the monk with the five roses (Figure 6). In these depictions the altar motif has been removed, and the image becomes an almost animated figure. It is revealing not only its sacred character (García Avilés 2007, p. 325; Wirth 1991, p. 161) but also its mediating role in the devotional context, that of achieving a closer and more accessible experience (Camille 2000, p. 243; Sansterre 2010, p. 165).

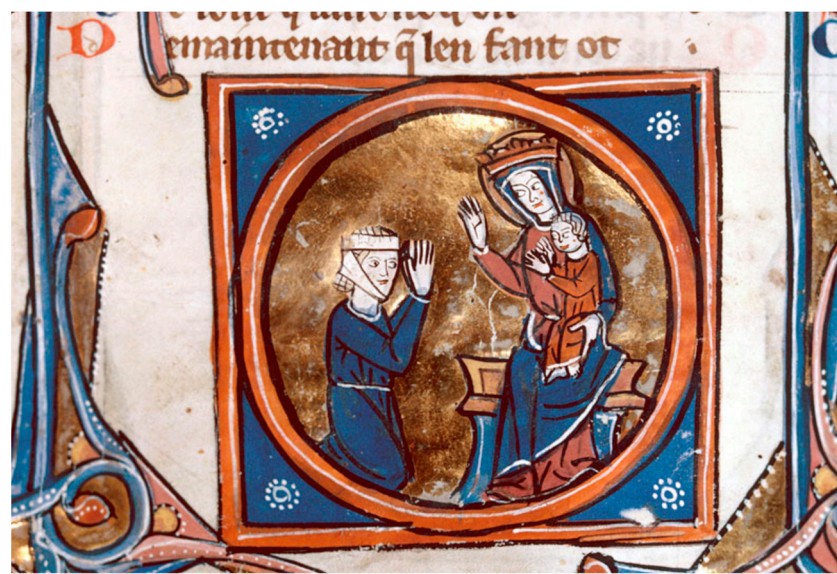

**Figure 4.** The noble woman of Rome. Gautier de Coinci, *Miracles de Nostre Dame*. Besançon, BM, MS 551, fol. 35r. Photo: Bibliothèque Municipale de Besançon.

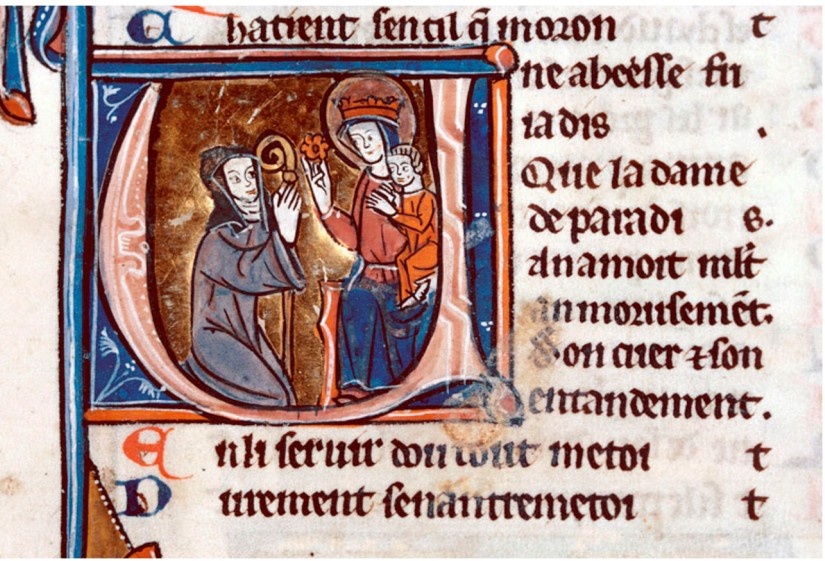

**Figure 5.** The pregnant abbess. Gautier de Coinci, *Miracles de Nostre Dame*. Besançon, BM, MS 551, fol. 41r. Photo: Bibliothèque Municipale de Besançon.

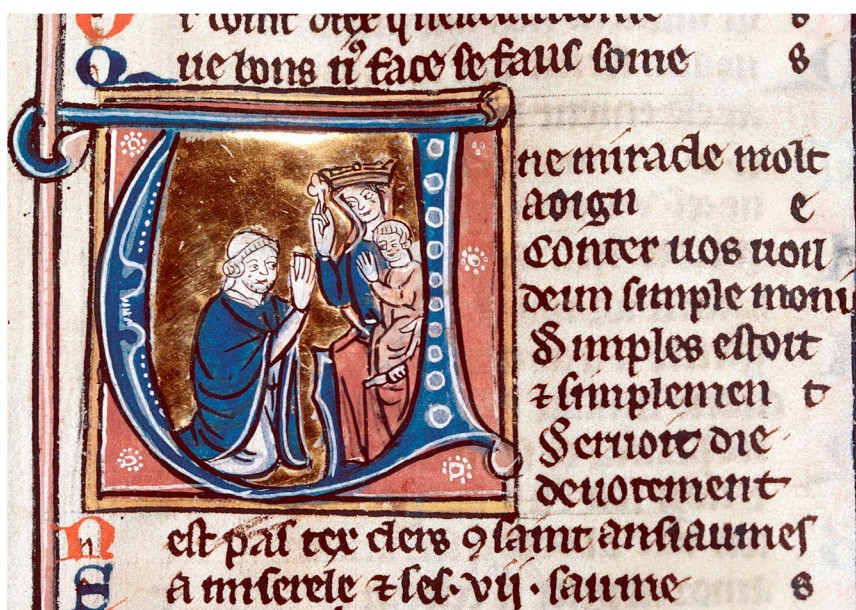

**Figure 6.** The monk with the five roses. Gautier de Coinci, *Miracles de Nostre Dame*. Besançon, BM, MS 551, fol. 46r. Photo: Bibliothèque Municipale de Besançon.

## 4. Seeing Is Believing: The Importance of the Gaze

One characteristic of Marian miracles is their universality: anyone can benefit from the Virgin's intervention, whatever their social status, sex, or creed (Cazelles 1978; Switten 2006). Thus, Gautier de Coinci records two miracles of conversion, that of the Jewish child and that of the Saracen. According to the account, the former attends the celebration of the Eucharist, where he contemplates an image of the Virgin:

> On the altar stood a beautifully carved image, its head covered with a mantle and with a child on its lap. The little Jew, when he stood before it, looked at it with attention, for he had found it very beautiful and gentle. In his heart he felt that he had never seen anything like it. It also seems to him that, instead of the priest, the image approaches him; it takes the host consecrated by the priest from the altar and makes him receive communion so sweetly that his whole heart overflows with joy[8].

The father, on learning of his son's habits, punishes him by throwing him into an oven, but he escapes unharmed thanks to the protection provided by Mary's mantle. In this case, the image allows him to identify the sacred model and then to provoke his conversion (Ahsmann 1930, pp. 92–96; Cuche 2012; Sansterre 2010, p. 163). The miniature at the head of the story depicts neither the punishment nor the salvation of the protagonist, but the moment when the statue gives him communion (Figure 7), which is unusual in the rest of the corpus (Murcia Nicolás 2012, pp. 179–80). This scene captures the vision in which he first contemplates her beauty and then "sees" how it is she, and not the priest, who gives him the consecrated host. The living image materializes the Virgin's function as the officiant of the Eucharist, an analogy based on the postulates of Bernard of Clairvaux (Angheben 2016, p. 166). She, as the personification of the Church, also shows true faith to the Jewish child through the animation of her statue. The real presence of the model allows the image to guide the vision towards the spiritual and the understanding of that which is invisible (Kessler 2021, p. 127). In contrast to idols, Christian representations stand as reliable proof of true faith and instruments of conversion (Geagea 1991; Roggema 2003; Russakoff 2019, p. 33). This anagogical process, based on contemplation, is also described in the Saracen story:



According to my books and my texts, a Saracen has an image with a representation of the Virgin Mary. I am not able to tell you, by my soul, neither where he found it, nor from where it came to him, but he held it very dearly, and kept it very carefully. The icon was painted richly with very rich colors. The Saracen held this icon in great reverence, and because it was so lovely and beautiful, he had gotten used to praying to it with great ardor: he adored it a least once a day on his knees with his hands joined ( . . . ) Thus, as God wanted, a day came that he came before this image. He looked at it for a very long time, and greatly in his mind wondered with astonishment if it could be true that this was the Mother of The Heavenly King, she whose image this was ( . . . ) While he was thinking in this way in the meantime and was reflecting in his mind, all at once from his image he saw two breasts appear and project, which were so glorious and so beautiful, so small and so perfectly made, as if just then they had been pulled out from the chest of a girl. As if from a fountain, he saw clear oil flow and come out of it. The Mother of God, the merciful, made this miracle happen, to extract him from his impiety, for he had much honoured and long protected her image[9].

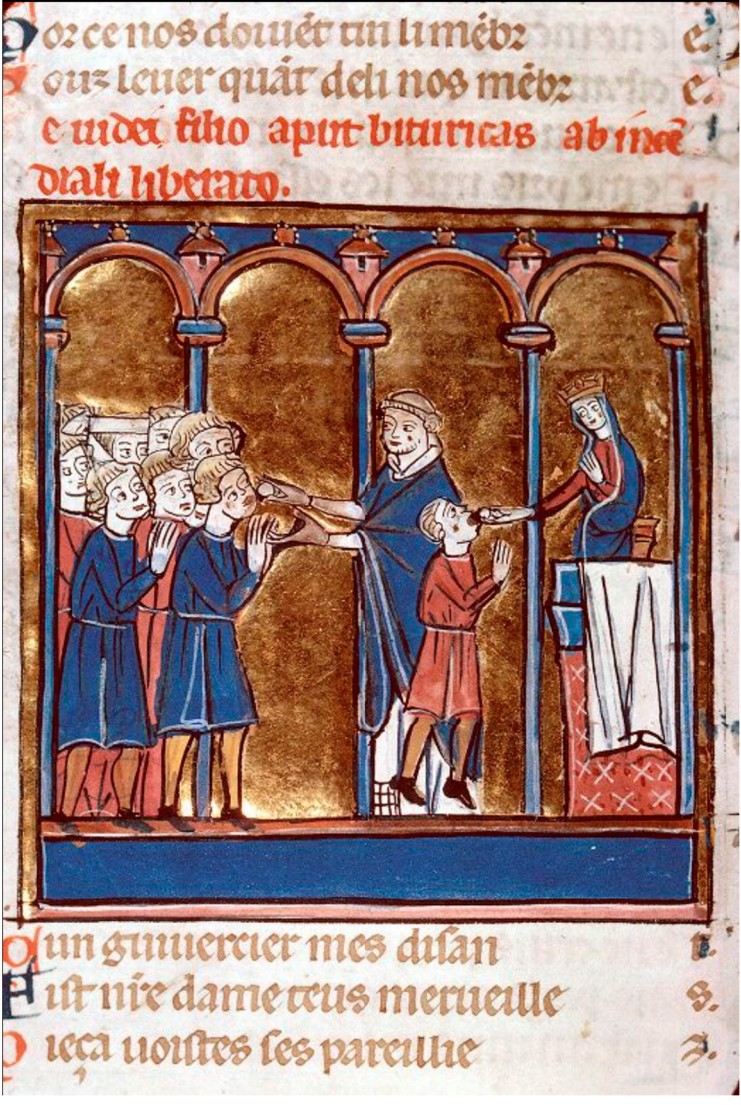

**Figure 7.** The Jewish child. Gautier de Coinci, *Miracles de Nostre Dame*. Besançon, BM, MS 551, fol. 31v. Photo: Bibliothèque Municipale de Besançon.

The description equates the beauty of the Virgin with that of her image, which is manifested when she is incarnated in her to bring about conversion (Sansterre 2010, pp. 161–62; Smith 2006). Thus, in the miniature, we see the image standing with her tunic open, showing her breast (Russakoff 2019, p. 31). As we have noted in the history of the desecrated icon, the illustrator represents a statue and not a painted image (Figure 8). This choice was made because the sculptural format favored a closer approach by the faithful (Bynum 2011, p. 70; Jung 2010, p. 215), but also because the miraculous transformation of the material is a way of moving the Saracen, who ends up becoming a Christian.

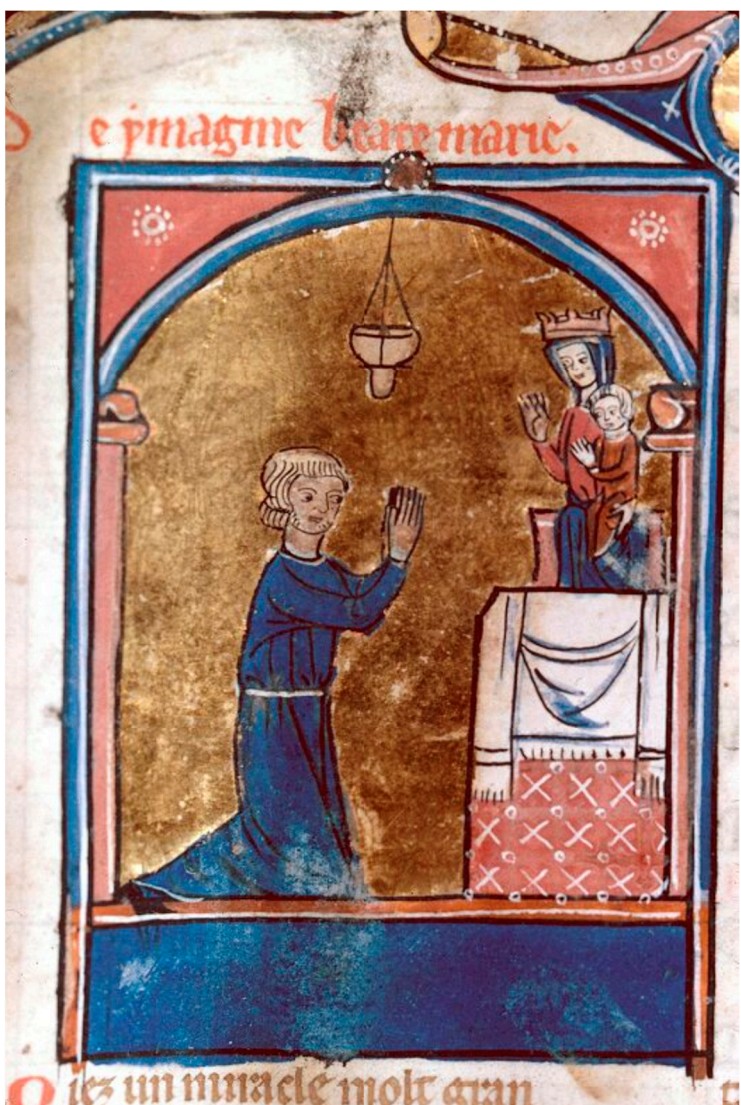

**Figure 8.** The conversion of Saracen. Gautier de Coinci, *Miracles de Nostre Dame*. Besançon, BM, MS 551, fol. 58r. Photo: Bibliothèque Municipale de Besançon.

These conversion stories focus on the gaze as the trigger for miraculous animation, in a contemplative context that, while beginning with a sensual appreciation, ends in a spiritual vision. This phenomenon was linked to the affective component that images, especially the Crucified One, could arouse (Sansterre 1998; Palazzo 2010; Schmitt 1994). However, the prominence given to the maternal and protective character of the Virgin means that these qualities are reflected in her depictions, which are described as beautiful and gentle (Oakes 2008; Rubin 2009b; Saxon 2006). In both miniatures, the living image is an agent of conversion which, beyond showing the relationship with its model, mediates the vision

towards an essential component of devotion: to believe, it is necessary to be able to see (Camille 1985; Hahn 2006).

## 5. The Devotee in Front of the Living Image

The examples examined so far are related to the text, despite the liberties taken by the illustrators to show the miraculous power of the images. However, at the beginning of the manuscript we find an exceptional illustration, which accompanies the prologue to the first book in the collection. In a historiated initial, Gautier de Coinci kneels in front of a living image of the Virgin (Figure 9). We have no textual support, since, throughout the prologue, the author explains the motivations for his work, from his interest in instructing potential readers to extolling Mary's qualities as a mediator (Benoit 2007; Montoya Martínez 1979). This living image therefore has no context; it has been freely added by the illustrator, who interprets Gautier de Coinci's fervor with devotion to Marian images.

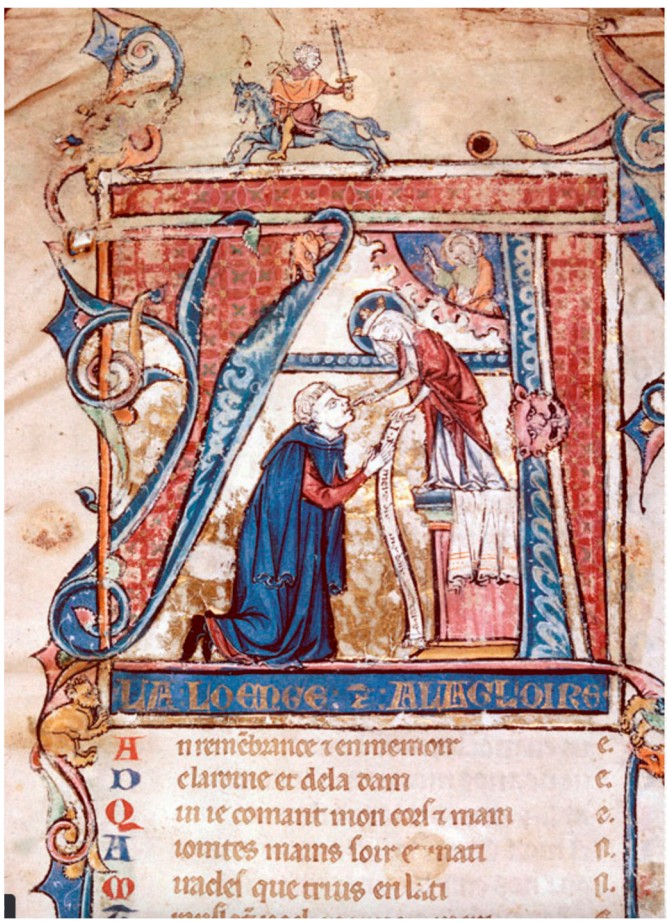

**Figure 9.** Gautier de Coinci in front of a living image. Gautier de Coinci, *Miracles de Nostre Dame*. Besançon, BM, MS 551, fol. 1r. Photo: Bibliothèque Municipale de Besançon.

His composition includes all the assumptions mentioned above. Firstly, the author is kneeling in front of an altar, a practice that he himself encourages in his accounts and which connects us with the idea of Byzantine transit (García Avilés 2007; Murcia Nicolás 2012, 2017). This gesture within a scene of private devotion refers us to the new liturgical connotations in which, like the Eucharistic substances, sculptures were the most frequently animated objects, an activation intended to produce a specific spiritual effect on the spectator (Palazzo 2020, pp. 116–18). This transformation is reinforced by the presence of the nimbus crowning the image. However, Gautier de Coinci not only kneels, he also prays, "Mother of God, have mercy", which is inscribed on the cartouche he holds in his hands. These visible words are in turn taken up by the image because prayer is a direct and

effective means of communication, rising from the faithful to the object of their devotion (Debiais 2017). Finally, we have the gaze, which not only contemplates how the image has come to life but also Christ in the heavenly sphere, which opens in the upper corner. The material component of the sacred image acts in the first instance by making the invisible visible, and then leads the gaze to its spiritual counterpart (Kessler 2021, pp. 148–49).

This miniature has a double interpretation. Firstly, the illustrator shows the value of the devotion given to Marian images and how these gestures, words, and gazes can activate them to come to life. Their relationship with the person they represent makes the Virgin respond by animating her statue, which touches Gautier de Coinci's mouth as a sign of gratitude. Secondly, however, the living image appears here as a link in two directions: on the one hand, it allows for the manifestation of Mary in the earthly world, and on the other, it is an ascending path for the devotee, who, after contemplating the prodigy worked by the *virtus sancta*, can contemplate and understand the divine. If the Virgin was placed as the closest mediator to Christ, the animated statue in this miniature exemplifies how Marian representations had assumed the qualities of their model.

## 6. Conclusions

The set of miniatures examined reflects the importance that the image had acquired in devotional and religious experience. The legitimization of its sacred character and the acceptance of its miraculous qualities generated a new visual culture, intended to show the close union they had with their model. In the case of the Virgin, her miracles become an extensive display of the roles they had acquired, as in Les Miracles de Nostre Dame. Gautier de Coinci stresses both Mary's mediating power and the need to honor her images through gestures, greetings, offerings, and prayers. His work, which is widely distributed, also shows us the new visual culture and all the iconographic resources forged around images as objects of worship, mediators, or agents of the miraculous.

Of the copies produced in the 13th century, the Besançon manuscript 551 is the most innovative in its representations. Despite the problems of its chronology and the flaws in its production, its illustrators explicitly depicted the qualities of the Marian images. In fact, it is the codex with the most living images in the entire corpus. Although some of them have a textual reference, the moments chosen from the narrative demonstrate the illustrators' interest in showing this virtue and no other. Thus, in the miracle of the desecrated icon, they interpret that it is the image and not the Virgin which punishes the Jew, and, furthermore, they opt for a sculptural format, more plausible to the reader's eyes. Moreover, the living attitudes appear for multiple reasons, from a young man's promise of love, the genuflection and greeting of a nun, to the gaze of a Jewish child. All these devotional practices are brought together in the first miniature of the manuscript, which corresponds to the prologue, where Gautier de Coinci kneels and says a prayer to the image he contemplates on an altar, which comes to life as a sign of gratitude, while above them the celestial sphere is contemplated.

In Les Miracles de Nostre Dame, the Virgin is present and materialized in her image, breaking the strict relationship between the model and her effigy and equating herself with her own representation—an idea that is not only narrated but also shown. The perception that matter could be modified by divine action and the Virgin's own close nature encouraged a vision of her images which, under the protection of the miracle, ended up becoming substitutes for their model. Examples such as manuscript 551 demonstrate the impact that conceptions of the sacred image had, but also set a precedent in medieval imaginary. As Michael Camille noted, "the collection of Gautier de Coinci, the most popular of the Marian miracles, although it takes up earlier Latin accounts, had a fundamental use in the new visual culture of the 13th century" (Camille 2000, p. 243).

**Funding:** This research was funded by Agencia Estatal de Investigación (Spain) for the project "PID2021-122593NB-I00. La experiencia de las imágenes (4): la recepción de la Antigüedad".

**Institutional Review Board Statement:** Not applicable.

**Informed Consent Statement:** Not applicable.

**Data Availability Statement:** Not applicable.

**Conflicts of Interest:** The author declares no conflict of interest.

## Notes

[1] Car en tant liuz fait la Dieu mere tant myracle et tante merveille touz li mondes s'en esmerville. I Pr 1, 54–56. Gautier de Coinci. *Miracles de Nostre Dame*. Koenig, vol. 1: 4.

[2] Un jor jooit une grant flote de clerçonciaus a la pelote devant les portaus de l'eglyse ou cele ymage estoit assise ( . . . ) Quequ'il pensoit en son corage, regardez s'est, se voit l'ymage, qui toute estoit freche et novele. Quant l'a veüe si tres bele, devant li s'est agenoilliez; devotement a ielz moilliez l'a enclinee et saluee. En peu de tans li fu muee la volontés de son corage. "Dame, fait il, tout mon aage d'or en avant te servirai, car onques mais ne remirai Dame, meschine ne pucele qui tant me fust plaisans et bele. Tu iez plus bele et plus plaisans que cele n'est cent mile tans qui cest anel m'avoit doné. Je li avoie abandoné tot mon corgae et tot mon cuer, mais por t'amor veil jeter puer li et s'amor et ses joialz. Cest anel ci, qui mout est biaus, te veil doner par fine amor par tel convent que ja nul jor n'arai mais amie ne fame se toit non, bele douce dame". L'anel qu'il tint bouta luez droit ou doit l'ymage, qu'ot tot droit. L'ymage tost isnelement ploia son doit si durement nus hom ne l'en poïst retraire s'il ne vossit l'anel desfaire. I Mir 21, 19–22, 33–634. Gautier de Coinci. *Miracles de Nostre Dame*. Koenig, vol. 2: 198–99.

[3] "Ce n'est mie, fait ele, drois ne loiautez que tu me fais; laidement t'iez vers moi mesfais. Vois ci l'anel a ta meschine, que me donas par amor fine et se disoiez que cent tans ere plus bele et plus plaisanz que pucele que tu seüsses. Loial amie en moi eüsses se ne m'eüssez deguerpie. La rose laisses por l'ortie et l'aiglentier por le seüz. Chetiz! Tu iez si deceüs que le fruit laissez por la fuelle, la lamproie por la suetuelle; por le venim et por le fiel laissez la ree et le doz miel" Li clers, qui mout s'esmervilla de l'avisïon, s'esvilla. Esbahis est en son corage. Lez lui cuide trover l'ymage. I Mir 21, 116–136. Gautier de Coinci. *Miracles de Nostre Dame*. Koenig, vol. 2: 201–2.

[4] Pres de lui en une fenestre garda et vit une tavlete ou painte avoit une ymagete a la samblance Nostre Dame. "Di moi, fait il, di moi, par t'ame, ceste ymage de cui est ele?" "Ele est, fait il, de la pucele qui tant fu pure, nete et monde que li sires de tot le monde humanité prist en ses flans". Au gïu boli toz li sans quant ul oï parler de li. "Aeures tu, fait il, celi que ne daignome nes nomer? On te devroit voir assomer ou acorer com une vache! Un viez piler ou une estache tout ausi bien puez honorer et encliner et aourer comme celi dont tu me contes. Fi! Fait li chienz, trop est grant hontes, trop grans viltance, trop grans diex quant nus hom croit que li gran Diex fust nez de cele marïole. Il en est mais tex carïole n'i a mostier ne mosteret ou il n'en ait ou sis ou set. Ains mais si grant honte n'avint!". I Mir 13, 16–43. Gautier de Coinci. *Miracles de Nostre Dame*. Koenig, vol. 2: 102.

[5] La mere Dieu, cui fu l'ymage, ne volt souffrir ce grant outrage. Cruelment et tost li meri, car paissïons luez le feri, se li sailli la langue fors. L'ame enporterent et le cors tout maintenent li anemi. I Mir 13, 47–53. Gautier de Coinci. *Miracles de Nostre Dame*. Koenig, vol. 2: 103.

[6] Quant vient la nuit de l'assamblee, fors del dortoir s'en est emblee mout coiement la damoisele. Droit par mi outre une chappele de Nostre Dame estoit sa voie. Ses dras escorce, si s'avoie vers la chappele isnelement. Batant sa coupe doucement ensi com l'avoit en usage, s'agenoille devant l'ymage. Quant humelement l'a salüee, isnelement s'est relevee. A l'uis en vient et passer cuide, mais l'ymage son estal wide, a l'uis en vient, plus n'i atent, ses bras en crois devant li tent. Grant piece i est ne se remuet si que cele passe ne puet ( . . . ) Et la nonne ne dormi mie, mais dou dortoir s'en ravala, vers la chappele droit ala, car n'i avoit nule autre voie. Mout humelement ses genolz ploie quant ele vint devant l'autel, et l'ymage refist autel com ele eut fait premierement. Esbahie est mout durement quant emmi l'uis revoit l'ymage, qui li devee le passage. Ses bras estent devant son vis si qu'il li samble et est avis que dire doie: "Bele amie, par ci ne passeras tu mie". I Mir 43, 93–100, 148–162. Gautier de Coinci. *Miracles de Nostre Dame*. Koenig, vol. 3: 194–97.

[7] Thomas Aquinas, *Summa Theologica* Pt. III, Q. 25, a. 3–4.

[8] Une ymage eut desor l'autel qui mout estoit de bele taille, deseur son chief une toaille, un enfançon en son devant. Li giuetiaus, quant vint devant, la regarda par grant entente, car mout li sambla bele et gente. Ses cuers bien li dist et revele qu'ainc mais ne vit chose tant bele. Avis li est en son corage qu'en liu del prestre vient l'ymage; desuer l'autel prise a l'oblee que li prestres avoit sacree; si doucement l'en commenie que toz li cuers l'en rasassie. I Mir 12, 20–34. Gautier de Coinci. *Miracles de Nostre Dame*. Koenig, vol. 2: 95–96.

[9] Ce dist mes livres et ma page c'uns sarrasins ot une ymage a la samblance Nostre Dame. A dire ne vos sai, par m'ame, ou la trova ne dont li vint, mais en mout grant chierté la tint et mout la garda netement. De riches colors richement painte estoit en une tavlete. Li sarrasinz cele ymagete avoit en mout grant reverence et aüssez s'estoi en ce, por ce que tant ert bele et gente, que chascun jor par fine rente l'aoroit une fois au mains a genolz et a jointes mains ( . . . ) Si com Diex volt, un jor avint que devant cele ymage vint. Mout longuement l'a regardee et durement en sa pensse se merveille se voirs puet estre que mere fust au roi celeste cele dont estoit cele ymage ( . . . ) Quesque penssoit en tel maniere, une eure avant et autre arriere, et devisoit en son corage, tot maintenant de cele ymage voit naistre et sordre deus mameles si glorïeuses et si beles, si petites et si bien faites con se luez droit les eüst traites fors de son saim une pucele. Ausi com d'une fontenele cler oile en voit sordre et venir. Cest myracle fist

avenir la mere Dieu, la debonaire, por lui de mescreance traire, car il avoit mout honoree s'ymage et longuement gardee. I Mir 32, 3–18, 25–31, 55–70. Gautier de Coinci. Miracles de Nostre Dame. Koenig, vol. 3: 23–25.

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
