# Peer review of "Living Images and Marian Devotion: Words, Gestures, and Gazes"

_religions, doi:10.3390/rel14050623_

Round 1

Reviewer 1 Report

The article meets the methodological requirements that may be required in a research work.

The paper is well structured, clearly states its objectives and the arguments presented are coherent and well articulated. Although, perhaps, the section of the conclusions could be developed a little more, because its consequences are very interesting from the philosophical and moral point of view.

Author Response

Dear Reviewer 1,

I have revised the wording to make the conclusion clearer. I thank you for your suggestion, which will improve the exposition of my research.

Reviewer 2 Report

I would recommend providing the translations of the quotations in the main text and the original text in a footnote.

The main question is the study of living images and relating them to those that speak of their religious experiences. This study provides a new approach because images usually are related to other kinds of discourses. For these reasons, this paper is interesting and relevant.
It’s original because it doesn’t study images only iconographically, the author also pays attention to the function of images in religious experiences. Studying images from religious experience provides a new point of view that unifies: theological, devotional, visual, and experimental approaches. In addition, the paper provides a new point of view considering the Virgin’s images as sacred and miraculous. This consideration shows a new visual culture in which images need to be honored by different practices. The author exemplifies Les Miracles de Nostre Dame and its images and texts.
The paper is well-structured and easy to read.
The conclusions perfectly answer the purposes of the paper.  

Author Response

Dear Reviewer 2

I have followed your recommendation and cite the original text in the footnotes. I thank you for your suggestion, which will improve the presentation of my research.

Reviewer 3 Report

The abstract says "we will focus our attention"; I don't know if the article has multiple authors (if not, this archaism should be corrected), but it reads as though it does.  The writing in Section 1 (Introduction) is often awkward to the point of opacity (e.g. "It has been necessary to clarify the debate regarding the semantic and ontological meaning of the images, which arose in the first centuries, and which ended up recognizing the possible uses they could have, while also relegating them to a mainly didactic function.").  It's not clear whether some of the awkwardness stems from a language barrier or if the author has had difficulty in lucidly distilling the thesis (e.g. "The approach to this problem from the religious experience brings a new perspective to the visual culture surrounding the  images, allowing us to understand their construction from all points of view.")  More specificity of what's to come in the article, including reference to illustrations, should be included in the Introduction.  

Sections 2-4 are clear and engaging.  Section 5 is very clunky and difficult to navigate (e.g. "We therefore find two levels of significance. Firstly, it summarizes the main devotional attitudes lent to the Marian images, and how these gestures, words and looks can activate them so that they come to life. In this way, the next level is established, in which the image has a close relationship with the personage it represents, which is why the Virgin responds by animating her statue that touches Gautier de Coinci's mouth, as a sign of gratitude. The living image is not a unidirectional link, which only allows the manifestation of the sacred personage, it is also an ascending path for the devotee who first contemplates the prodigy worked by the virtus sancta, and then has a higher vision of the divine.)

Parts of Section 6 (Conclusion) are also very difficult to work through; for one thing, there is occasionally a confusion of pronouns and antecedents (e.g. "The set of miniatures examined reflects the importance that the image had acquired in devotional and religious experience. The legitimization of their sacred character and the acceptance of their miraculous qualities generated a new visual culture, intended to show the close union they had with their model.") 

The phrase "visual culture" is awkwardly shoehorned into the Introduction and Conclusion and does nothing to enhance the import of the material covered. 

Author Response

Dear Reviewer 3,

Thank you for your comments on my article. I have removed the translation errors and made changes to the sentences you have pointed out to me in sections 5 and 6.

Once again, thank you very much for your review, which improves the final version of my research.

Round 2

Reviewer 3 Report

This article is fine for publication now.